# The impact of policy restrictions and mobility changes on excess mortality during the COVID-19 pandemic in The Netherlands, 2020–2022

Dimiter Toshkov[1]*, Camila Caram-Deelder[2], Brendan Carroll[1], Frits Rosendaal[2]

**1** Institute of Public Administration, Leiden University, Leiden, the Netherlands, **2** Department of Clinical Epidemiology, Leiden University Medical Center, Leiden, the Netherlands

* d.d.toshkov@fgga.leidenuniv.nl

## Abstract

We analyzed the impact of the COVID-19 policy restrictions on mobility patterns and excess mortality at the regional level in The Netherlands between 2020 and 2022. Our analysis combines data on public policies, mobility patterns from the Google Mobility Reports, officially registered COVID-19 cases and deaths, and new region-specific measures of excess mortality over a relatively long time period extending beyond the first wave of the pandemic. We modeled the relationships of these so that policy responds to information about the pandemic; mobility reacts both to information about the pandemic and to policy; the number of COVID-19 cases is influenced by changes in mobility and policy; and excess mortality is affected directly by the policy restrictions and indirectly via the impact of policy on mobility. The results confirm that the stringency of policy restrictions increased with the number and growth rates of COVID-19 cases and deaths. Mobility, as reflected in presence in public places (transport hubs, groceries, retail, work), decreased while presence at residential locations increased in response to stricter policies and higher COVID-19 case and death counts in preceding weeks. The number of new COVID-19 cases declined when stricter policy restrictions were enacted and with reduced presence in public places (following a two-week lag). Excess mortality decreased with stricter policy restrictions (with a five-week lag) and, to a lesser extent, with reduced presence in public places and increased presence in places of residence. Importantly, the effects of policy restrictions and mobility diminished with consecutive COVID-19 waves. Overall, the evidence shows that policy restrictions were effective in limiting the spread of the pandemic and in saving lives. While policies influenced mobility patterns, the policy impact was not fully mediated by mobility changes.

**Data availability statement:** All relevant data for this study are publicly available from the Zenodo repository (https://doi.org/10.5281/zenodo.17952217).

**Funding:** FR and CCD received funding from the Dutch organization for knowledge and innovation in health, healthcare and well-being (ZomMW): No 104.302.522.100.07 and 104.302.522.200.03. The funder did not play any role in the study design, data collection and analysis, decision to publish, or preparation of the manuscript.

**Competing interests:** The authors have declared that no competing interests exist.

## Introduction

Starting in early 2020, governments around the world enacted restrictive policies (non-pharmaceutical interventions) with the aim to curb and slow down the spread of the COVID-19 pandemic. These policies included physical distancing, face mask requirements, school closures, restrictions on large gatherings, recommendations to work from home, bans on use of public spaces such as parks, reduced opening hours of shopping centres, and – ultimately – lockdowns and curfews [1]. These policies imposed significant costs on societies, including limits on personal liberties and financial losses for businesses. At the same time, the restrictions may have saved lives by containing the spread of the coronavirus, protecting the capacity of hospitals, and limiting the reach of COVID-19. It is therefore imperative to assess their impact on excess mortality, as one crucial indicator of the toll of the pandemic.

There are a significant number of studies that have attempted to assess the impact of the policy restrictions on health outcomes (i.a. [2–12]). These studies examine various interventions including mask mandates (finding 10–47% mortality reductions; [2]), school closures (58–72% reductions in mortality when implemented early; [3,7]), mobility restrictions (2.9% increased mortality per 1% mobility increase; [4]), and comprehensive lockdown measures (with significant effects on various indicators depending on the set of measures analysed; [5,6,10–12]). Some research challenges these findings, reporting no mortality differences from shelter-in-place orders [8] or highlighting state-level heterogeneity in policy effectiveness [9]. On balance, existing research finds strong evidence that restrictive measures reduced the number of infections and the number of deaths related to COVID-19, although there is heterogeneity in the effects of different policies and context-specificity of these effects (for systematic reviews, see [13–15]).

While many studies address the question of COVID-19 policy impact, the vast majority of these studies focus on the first wave of the pandemic and the government responses to it. Yet, the effectiveness of measures may have varied over the different waves of the pandemic. The effectiveness may have varied due to different levels of compliance (e.g., 'pandemic fatigue') and – as a result – different effects on the mediator 'mobility'. Policy effects might vary because of changes in the mix of policy measures over time as well. Furthermore, even studies that evaluated the impact on the death toll of the pandemic often focused on the number of COVID-19 related deaths as reported by the governments, while excess mortality is a more appropriate measure [16–21]. This is because the number of deaths registered as caused by COVID-19 does not accurately represent the actual death toll of the pandemic due to inconsistent testing availability, deaths occurring outside healthcare settings, and indirect effects such as disrupted medical care for other conditions, and because of regional and temporal variation in the decisions leading to classifying a death as COVID-19 related or not. There are also a limited number of studies that explore within-country variation in excess mortality, while having regional-level mobility data provides additional opportunities to assess the indirect policy impact mediated by changes in mobility.

In this article, we assess the impact of the policy measures by examining the relationships between policy, changes in patterns of mobility over time and across regions within a country, changes in registered cases and differences between successive pandemic waves (both of which may affect the level of policy restrictions

and mobility directly) and, ultimately, excess mortality. Our analysis is grounded in a model in which changes in mobility (people spending more time at home and less time in public spaces such as transport hubs, shopping centres and offices) provide an important but not the only mechanism through which the public policies could have affected COVID-19-induced excess mortality. While endogenous policy adoption precludes definitive causal identification, we provide policy-relevant evidence by estimating conditional associations that control for the pandemic information available to decision-makers. Specifically, we address what mortality reductions accompanied policy restrictions given the conditions that prompted them, that is, the practical question facing policymakers during the pandemic. The main research questions are (1) what was the effect of policy restrictions on mobility, (2) what was the effect of changes in mobility on excess mortality, (3) what was the effect of restrictive policy measures on excess mortality overall and specifically via reducing mobility?

We study these questions focusing on The Netherlands in the period from 2020 to 2022, with variation over time and between the 12 regional units (provinces) in the country. During this period, various policy restrictions were imposed, modified and eventually lifted. While there was limited variation in the presence or absence of restrictive policies *across* provinces in the country, compliance with the policies, as reflected in changes in mobility patterns, may have differed. This variation allows us to estimate the indirect effect of the policy restrictions, as recommended in recent work [22].

The causal effect of the policies is difficult to assess because the imposition of restrictions followed the course and severity of the pandemic: restrictive policies were more likely to be adopted when COVID-19 cases surged, which in itself leads to increases in mortality. Furthermore, there is not much variation in the timing and nature of restrictions imposed *within* most individual countries (and only to some extent across countries). These associations and lack of variation, as well as related concerns about confounding, are major challenges for the identification of the causal impact of the COVID-19 policy measures.

Various research designs and methodologies have been employed to assess the effects of the policies. For example, Méndez-Lizárraga et al. [4] use change-point analysis. Chernozhukov et al. apply econometric techniques to panel data from US states [2] and US counties [3]. Askitas et al. [5] use daily data from 175 countries to estimate the dynamic effects of different policy interventions.. Some studies focus only on the relationship between mobility changes and health outcomes (assessed with correlation coefficients) (e.g., [23]), others use changes in mobility as a proxy for policy (e.g., [4]), while the most sophisticated incorporate mobility in the models designed to assess the policy impact [2,3].

The main strengths of our approach compared to the existing literature are that (a) we use a highly disaggregated (weekly province-level) estimate of excess mortality, which provides a reliable and fine-grained measure of the death toll of the pandemic [16–20,24]; (b) we combine policy, mobility and COVID-19-related data so that we can assess simultaneously various direct and indirect paths of influence; and (c) we analyse an extended period of time that goes beyond the first wave of the pandemic, which is the focus of the vast majority of existing studies. This is important, as the effects of policy and mobility might have changed considerably over the different waves of the pandemic. In addition, in our statistical models we control for important potential confounders of these relationships, including demographic variables related to the population structure of the provinces, as well as data on the weather that might have affected both mobility patterns and mortality.

Our study found that policy stringency increased in response to higher numbers and growth rates of COVID-19 cases and deaths, with the impact of deaths increasing over the course of the pandemic. Stricter policy in turn reduced citizens' mobility, while deaths (and to a lesser degree cases) also had a direct negative effect on mobility. Policy restrictions and changes in mobility had significant effects on the number of cases after a lag of two weeks. Both affected significantly excess mortality as well, but collinearity makes it difficult to estimate both effects simultaneously. Importantly, our study showed that while the effects of policy restrictions and mobility weakened with successive waves of the pandemic, overall, the restrictions were effective in reducing mobility and in saving lives.

## Theoretical causal mechanisms

In this section we outline the theoretical model that we use to guide the empirical analysis. The model incorporates the following hypothesized relationships. First, as the pandemic unfolds, information about the state of the spread and impact of COVID-19 affects the likelihood that restrictive policy measures are adopted (and later on lifted). The most policy-relevant

information signals have been the number of new COVID-19 cases and deaths, as well as their growth rates. Policy stringency is expected to increase with a higher number and higher growth rates of COVID-19 cases and deaths, and to decrease when fewer cases and deaths are registered and the pandemic subdues.

Pandemic-related information might also impact mobility patterns directly, i.e., not as a result of restrictions, as citizens will tend to spend more time at home and avoid public places when higher numbers of COVID-19 cases and deaths are reported for fear of the virus. Mobility might also respond to the course of the pandemic via the path Information > Policy restrictions > Mobility changes. Many policy measures limited presence in public places by imposing physical distancing, school closures, bans on large gatherings, work-from-home recommendations, restricting access to parks and shopping centres and total societal lockdowns. We hypothesize that restrictive policy measures affect mobility patterns immediately when imposed without a lag (at the week level of aggregation).

Next, policy restrictions and changes in mobility affect the number of COVID-19 cases, but with a lag of $j$ weeks (for example, two weeks). Some of the policy-related reductions are expected to be exercised via changes in mobility. But it is also possible that restrictive policy measures affect the spread of the pandemic via alternative mechanisms, for instance improved ventilation in buildings and face mask requirements, or testing. Finally, changes in the number of COVID-19 cases lead to COVID-19 related deaths and excess mortality, with a lag of $k$ weeks (for example, three weeks).

As COVID-19 cases and growth affect policy (and mobility) and, in their turn, policy and mobility affect the number of new COVID-19, it would seem that our model contains a cyclic causal relationship, which presents a challenge for causal identification. But the temporal ordering and lag structure that we impose (mobility and policy react instantaneously to changes in the COVID-19 growth rate, but affect cases only after a lag of time) provides a solution to this challenge (cf. [2,3]). Our causal identification approach is valid if changes in the stringency of policy measures can be considered 'as random' conditional on the covariates included in the models, the most important of which are the variables capturing information about the state of the pandemic (COVID-19 cases, deaths and their growth rates). For a different approach, see [25] who attempt to learn the causal structure of this system of variables inductively via machine learning.

## Materials and methods

The study is based on an observational time-series cross-sectional design that exploits variation in the imposition of public policy measures over time and variation in the changes in mobility patterns over time *and* across spatial units. Hence, the unit of observation is a province in a week. The spatial unit is the province level in The Netherlands ($N_{provinces}$ = 12). The provinces differ significantly when it comes to population density, from in 187 people per km$^2$ in Drenthe to 1374 in South Holland (see the Supplementary material for details). The time scope of the analysis is between third week of February 2020 and the second week of October 2022, which is the last date of availability for the mobility data ($N_{weeks}$ = 140).

We translate our theoretical model in the following equations:

$$\text{Mobility}_{t,i} \sim \text{Information}_{t,i} + \text{Wave}_{t,} + \text{Policy}_{t,i} + \text{Temperature}_{t,i} + \text{Public holidays}_{t} + \text{Confounders}_{it} \quad (1)$$

$$\text{COVID} - 19\ \text{cases}_{t,i} \sim \text{Mobility}_{t-j,i} + \text{Policy}_{t-j,i} + \text{Wave}_{t,} + \text{Public holidays}_{t} + \text{Confounders}_{it} + \text{COVID} - 19\ \text{cases}_{t-1,i} \quad (2)$$

$$\text{Log}(\text{Excess mortality}_{t,i}) \sim \text{Mobility}_{t-k,i} + \text{Policy}_{t-k,i} + \text{Temperature extremes}_{t,i} + \text{Public holidays}_{t} + \text{Confounders}_{it} \quad (3)$$

In the equations above, $t$ is a week indicator and $i$ is a spatial unit indicator. $j$ is the indicator lag time for the measurement of number of cases and $k$ is the lag time for the policy. The effect of the public policy restrictions via mobility is given by the product of the coefficient of Policy in equation 1 and the coefficient of changes in Mobility in equation 2, for COVID-19 cases and in equation 3 for excess mortality.

We use linear regression to estimate the equations. Where appropriate, we add indicators for the provinces. In addition, we explore interactions of the main variables of interest with the COVID-19 pandemic wave and across the spatial units.

The variable 'Policy' is operationalized as the general policy stringency index constructed from the Oxford COVID-19 Government Response Tracker [26]. The variable is aggregated per week by using the mean for the week and is the same for all provinces. We rescale the variable from the original 0-to-100 range to the 0-to-10 range to improve coefficient readability of our results. The variable tracking pandemic-related 'Information' is operationalized as the natural logarithm (the log) of the number of new province-specific weekly COVID-19 cases and the number of new province-specific weekly COVID-19 related deaths. We use the seven-day moving average of the daily values of cases and deaths and then aggregate per week by taking the mean for the seven days of the week.

The variable 'Mobility' is operationalized as an index constructed from the Google Mobility Report, which tracks for each date the percentage change in presence at different categories of places in a province relative to a baseline period in January-February 2020 in the same province. We take the seven-day moving average of the daily values and then aggregate per week by taking the mean for the seven days of the week. We focus on mobility measures about five particular kinds of places (i.e., places of work, transport hubs, grocery shops and markets, retail and recreation, and residential places), as reported by the Google Mobility Report. We exclude the sixth available category – 'Parks' – because of uncertain theoretical expectations about presence in parks and nature areas in response to the pandemic and policy.

'Excess mortality' is operationalized as the number of deaths in a province in a week relative to the baseline in 2019 in the province and the week. We use Poisson regression analysis to estimate excess mortality in terms of incidence rate ratios (IRR) for each week 2020, 2021 and 2022 compared with the baseline year (2019). The models are based on individual-level register data for the entire Dutch population provided via a special arrangement by the CBS (Statistics Netherlands). The study was approved by the Scientific Committee of the Department of Clinical Epidemiology of the Leiden University Medical Center (protocol A0199) with a waiver of participant consent, because it used exclusively pre-existing, de-identified data, which the Dutch Statistics Office (CBS) is allowed to process by law. The measure adjusts for the composition of the population in the province in terms of sex, age, household income and immigration background, and for the calendar month as well. We take the log of the IRR. More details about the estimation and the mathematical models behind the estimates is provided in [21]. S3 and S4 Figs in S1 File shows the descriptive trends in these main variables of interest.

Our models also include some additional covariates that capture additional influences on mobility, COVID-19 cases and excess mortality. We define the pandemic waves as first (January-July 2020), second (August 2020-July 2021), and third (July 2021 onwards). Public holidays is a variable that is necessary to include because of the way mobility changes are measured (relative to a baseline in February). We would expect that this variable picks up the effect of reductions in mobility that are not related to COVID-19 or policy responses, but to other relevant events. The variable indexes the weeks of Christmas, New Year and Easter. The variables related to Temperature are operationalized in the following ways. In the models of Mobility and COVID-19 cases, we include a measure of the weekly average of the maximum daily temperature. In the models of excess mortality, we include measures that tracks whether the weekly average of the daily minimum temperature was below 0°C or the weekly average of the daily maximum temperature was above 25°C. When this was the case, we counted the 'excess' degrees below 0°C or above 25°C. This is done on the basis of the expectation that it is days with extreme positive or negative temperatures that might lead to excess mortality. All temperature measures are province-specific, taken at a weather station within each province.

The demographic covariates relate to the composition of the population of the province in a week. In particular, we track the share of old people (65+), the share of women, the share of low and low-middle income households and the share of 1st generation immigrants. The variable has some variation within provinces over time, but the extent of this variation is limited.

All models reported below are estimated using the R software environment for statistical computing and graphics using base R and the *fixest* package [27] for the models reported in S1 Table in S1 File. The figures of marginal effects are produced with the packages *modelsummary* [28] and *coefplot* [29].

## Results and discussion

### Part I: Mobility as a function of COVID-19 cases and policy restrictions

Table 1 presents five linear regression models that analyse changes in mobility. Each of the five models focuses on a different aspect of mobility. The dependent variable is the percentage change in the presence of people in particular types of places compared with a baseline period in the pre-pandemic period in early 2020. All five models include indicators at the level of the province, as well as control variables related to the demographic structure of the population in the province (which has some minor variation within-provinces over time as well).

The policy stringency index has statistically significant negative associations with presence in places of work, public transport hubs, grocery shops and markets, and retail and recreation; it has a positive association with the duration of presence in residential places. This implies that mobility responded to public policy, and people reduced their presence in work, transport and shopping places and stayed more at home, in accordance with the policy restrictions.

The effect of public policy comes on top of any direct responses of the population to the state of the pandemic, as captured by the information variables related to the number of cases and deaths.

The number of deaths has significant negative associations with mobility, so that in the aftermath of weeks with more COVID-19-related deaths people spent less time in places of work, public transport hubs, retail and recreation, and grocery shops and markets. People spent more time in residential places. The number of reported cases also has the expected effects, except on Places of work, where the effect is not significant, possibly due to collinearity with the number of deaths. Whether the number of deaths is log-transformed or not makes no substantive difference for these inferences. Similarly, using the number of deaths with lag 2 does not change the conclusions.

**Table 1. Changes in mobility as a function of policy stringency, COVID-19 cases and deaths, and additional covariates.**

| | Model 1a Places of work | Model 1b Public transport hubs | Model 1c Grocery shops and markets | Model 1d Retail and recreation | Model 1e Residential places |
|---|---|---|---|---|---|
| Policy stringency | −2.74 [−3.03, −2.46] p<0.01 *** | −3.14 [−3.50, −2.79] p<0.01 *** | −0.51 [−0.80, −0.22] p<0.01 *** | −3.99 [−4.44, −3.55] p<0.01 *** | 0.96 [0.89, 1.02] p<0.01 *** |
| COVID-19 cases (log, lag 1) | 0.05 [−0.33, 0.42] p=0.81 | −2.30 [−2.77, −1.83] p<0.01 *** | −0.96 [−1.34, −0.58] p<0.01 *** | −0.87 [−1.45, −0.29] p<0.01 ** | 0.41 [0.32, 0.49] p<0.01 *** |
| COVID-19 deaths (log, lag 1) | −1.39 [−1.86, −0.93] p<0.01 *** | −1.44 [−2.02, −0.85] p<0.01 *** | −0.61 [−1.09, −0.13] p=0.01 * | −2.98 [−3.70, −2.26] p<0.01 *** | 0.61 [0.51, 0.72] p<0.01 *** |
| Second wave | 4.50 [2.96, 6.03] p<0.01 *** | 11.28 [9.36, 13.20] p<0.01 *** | 4.86 [3.30, 6.42] p<0.01 *** | 7.88 [5.51, 10.25] p<0.01 *** | −3.37 [−3.72, −3.02] p<0.01 *** |
| Third wave | −3.56 [−5.93, −1.20] p<0.01 ** | 14.97 [12.01, 17.92] p<0.01 *** | 10.55 [8.15, 12.95] p<0.01 *** | 15.31 [11.66, 18.97] p<0.01 *** | −3.41 [−3.95, −2.87] p<0.01 *** |
| Public holidays | −10.51 [−12.07, −8.95] p<0.01 *** | −4.87 [−6.82, −2.93] p<0.01 *** | −5.45 [−7.04, −3.86] p<0.01 *** | −9.11 [−11.53, −6.69] p<0.01 *** | 1.82 [1.46, 2.17] p<0.01 *** |
| Weekly average of daily maximum temperature | −0.43 [−0.50, −0.36] p<0.01 *** | 0.34 [0.25, 0.42] p<0.01 *** | 0.38 [0.31, 0.46] p<0.01 *** | 1.25 [1.15, 1.36] p<0.01 *** | −0.08 [−0.10, −0.07] p<0.01 *** |
| Num.Obs. | 1626 | 1622 | 1630 | 1628 | 1632 |
| R2 Adj. | 0.519 | 0.798 | 0.664 | 0.803 | 0.859 |

The numbers show the unstandardized coefficients from linear regression models, which indicate the implied change on Mobility (defined as the percentage change in the presence of people compared with a baseline period in the pre-pandemic period in early 2020 in particular types of places) for a one-unit change in the covariate. The models include indicators at the province level (N=12), as well as controls for the demographic structure of the provinces (share of 65+, share of women, share of low-income households, share of 1st generation immigrants). 95% Confidence intervals are reported in the square brackets. Significance levels of p values: ***<0.001; **<0.01; *<0.05; +<0.10. The precise p values are printed when >0.01.

We also note that the variables for the weeks with big public holidays (Christmas and Easter) and the weekly average of the daily maximum temperature in the province have the expected associations with changes in mobility, with people spending more time at home and less time in other places during holidays and less time at home and at the office when it is warmer.

When we include interaction effects between the policy stringency index and the COVID-19 waves, we can see that the effects of policy on mobility are substantially attenuated during the second and the third wave, but remain significant and in the direction reported in Table 1. The results of these models are reported in S2 Table and illustrated in S2 Fig in S1 File.

These models also show that the number of COVID-19 cases has the expected effect only during the first wave, but not after (which might explain the overall lack of a consistent effect of this variable in Table 1). The number of deaths has the strongest effect in the expected direction during the second wave of the pandemic. We also explored possible differential effects of policy and COVID-19-related information on mobility across different provinces. The main results of these models with respect to the effects of policy stringency are presented in Fig 1.

The figure shows that the negative effect of policy stringency on mobility related to places of work was greatest in Limburg and Gelderland; with regard to transport hubs, it was greatest in Noord-Holland; with regard to grocery shop and markets, it was variable but greatest in Limburg and Zeeland; and with regard to places of residence, the positive effect of policy stringency was greatest in Zeeland, but the variation across provinces was minor.

## Part II: COVID-19 cases as a function of policy restrictions and mobility changes

Next, we look at the number of COVID-19 cases as a function of the lagged value of the number of COVID-19 cases, the lagged value of policy stringency, and the lagged values of the changes in mobility. All linear regression models reported in Table 2 also include province indicators and demographic controls for the population structure, in addition to the indicators for the pandemic wave, public holidays and a variable tracking the weekly average of the daily maximum temperature in the province.

As expected, the number of COVID-19 cases is strongly auto-correlated, as indicated by the significant lagged value of this variable and the adjusted R-squared for the models. More importantly, the lagged value of the policy stringency index has negative association with the number of COVID-19 cases, implying that the policy restrictions worked to reduce the spread of the pandemic.

The variables tracking different type of mobility changes also have significant effects (with one exception). More time at work, transport and grocery places (or, equivalently, smaller reductions in presence at such places compared to the baseline) is associated with more COVID-19 cases two weeks after. More presence at home (residential places) has a negative association with the number of COVID-19 cases. We obtain substantively the same results when we model the growth rate in cases rather than the level.

When we include interactions between the effects of policy stringency and mobility with COVID-19 waves, we find evidence that the effects varied over the course the pandemic (see S3 Table in S1 File). Fig 2 illustrates the results with respect to policy. The negative effect of policy restrictions was strongest during the second wave, while it was weakest – and not estimated precisely to be different from zero – during the third wave. Mobility changes related to work and transport places had significant effects mostly during the first wave. There is not much variation in the effect of policy across provinces.

We should note that the imposition and lifting of policy restrictions itself is related, as expected, to the number of COVID-19 cases and deaths and their growth rates in the preceding week (see S1 Table and S1 Fig for details in S1 File).

## Part III: Excess mortality as a function of policy restrictions and mobility changes

Lastly, we look at excess mortality per week in each province as a function of the policy stringency index, changes in mobility and province indicators and demographic covariates (Table 3). Policy stringency (lagged with 5 weeks) has

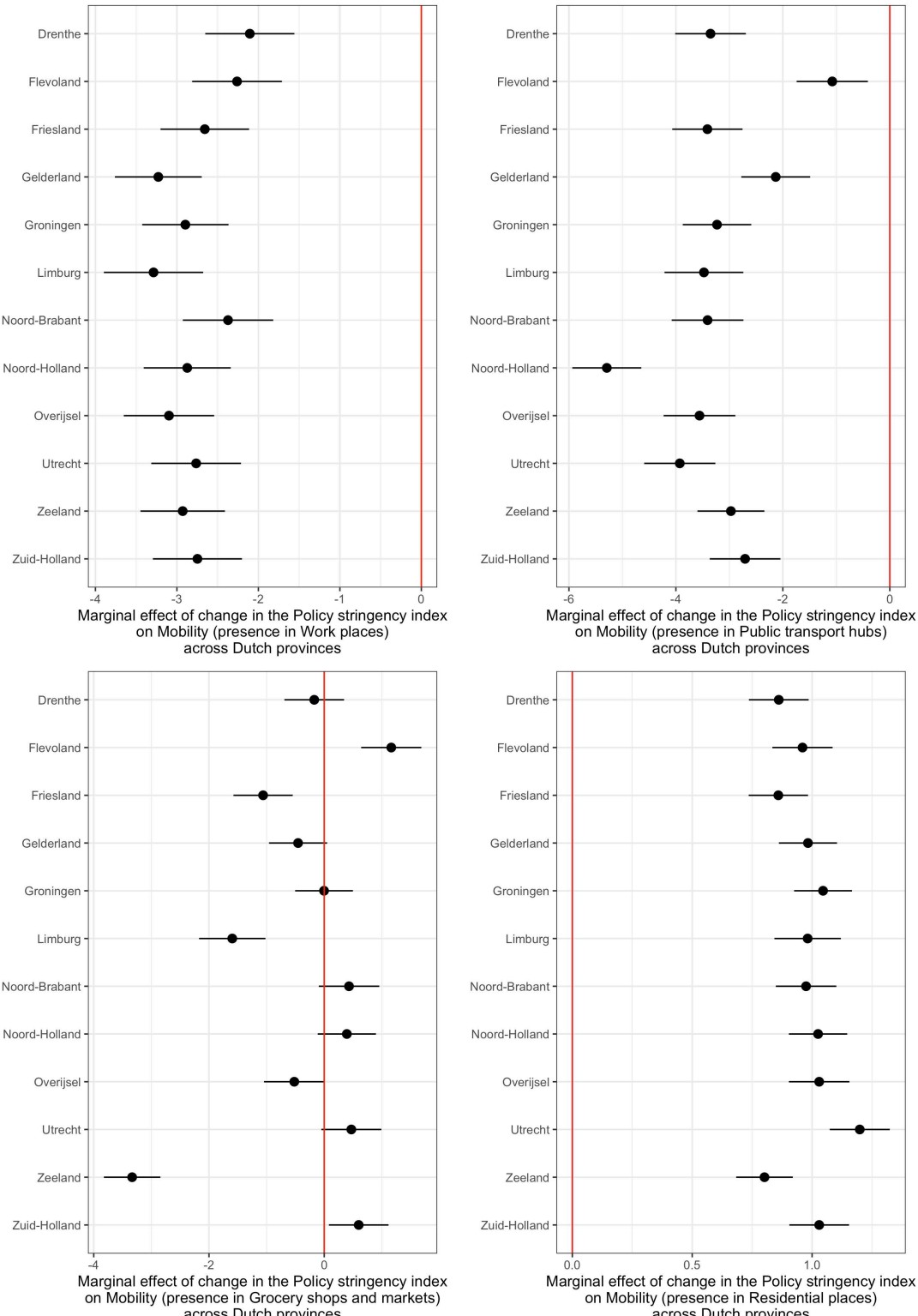

**Fig 1. Marginal effects of changes in the Policy stringency index on changes in mobility for four types of places across provinces in The Netherlands, 2020-2022.** The figure shows the predicted marginal effects ((point estimates and 95% confidence intervals) of one-point increases in the Policy stringency index on Mobility (defined as the percentage change in the presence of people compared with a baseline period in the pre-pandemic

period in early 2020) in four types of places (Work places, Public transport hubs, Grocery shops and markets, and Residential places) depicted in the four panels of the figures, in each of the 12 provinces in The Netherlands. Results based on variations of the models reported in Table 2 with added interactions of policy stringency and province indicators.

**Table 2. Number of registered COVID-19 cases (logged) as a function of policy stringency, changes in mobility and additional covariates.**

| | Model 2a COVID-19 cases (log) | Model 2b COVID-19 cases (log) | Model 2c COVID-19 cases (log) | Model 2d COVID-19 cases (log) | Model 2e COVID-19 cases (log) | Model 2f COVID-19 cases (log) |
|---|---|---|---|---|---|---|
| Lagged COVID-19 cases (log) | 0.83 [0.81, 0.85] p=<0.01 *** | 0.83 [0.81, 0.85] p=<0.01 *** | 0.84 [0.82, 0.86] p=<0.01 *** | 0.83 [0.81, 0.85] p=<0.01 *** | 0.83 [0.80, 0.85] p=<0.01 *** | 0.84 [0.82, 0.86] p=<0.01 *** |
| Public holidays | −0.04 [−0.14, 0.06] p=0.42 | −0.03 [−0.13, 0.06] p=0.52 | −0.02 [−0.12, 0.07] p=0.62 | −0.04 [−0.14, 0.06] p=0.42 | −0.04 [−0.14, 0.06] p=0.40 | −0.02 [−0.12, 0.07] p=0.62 |
| Weekly average of daily maximum temperature | −0.04 [−0.04, −0.03] p=<0.01 *** | −0.03 [−0.04, −0.03] p=<0.01 *** | −0.04 [−0.04, −0.03] p=<0.01 *** | −0.04 [−0.04, −0.03] p=<0.01 *** | −0.03 [−0.04, −0.03] p=<0.01 *** | −0.04 [−0.04, −0.03] p=<0.01 *** |
| Second wave | 0.47 [0.38, 0.56] p=<0.01 *** | 0.44 [0.35, 0.53] p=<0.01 *** | 0.40 [0.31, 0.50] p=<0.01 *** | 0.45 [0.36, 0.54] p=<0.01 *** | 0.48 [0.38, 0.58] p=<0.01 *** | 0.37 [0.27, 0.47] p=<0.01 *** |
| Third wave | 0.47 [0.33, 0.61] p=<0.01 *** | 0.49 [0.35, 0.63] p=<0.01 *** | 0.39 [0.25, 0.53] p=<0.01 *** | 0.43 [0.29, 0.58] p=<0.01 *** | 0.48 [0.33, 0.64] p=<0.01 *** | 0.37 [0.22, 0.52] p=<0.01 *** |
| Policy stringency (lag 2) | −0.07 [−0.08, −0.05] p=<0.01 *** | −0.05 [−0.06, −0.03] p=<0.01 *** | −0.05 [−0.07, −0.03] p=<0.01 *** | −0.06 [−0.08, −0.05] p=<0.01 *** | −0.07 [−0.09, −0.05] p=<0.01 *** | −0.04 [−0.06, −0.02] p=<0.01 *** |
| Work (lag 2) | | 0.01 [0.00, 0.01] p=<0.01 *** | | | | |
| Transport (lag 2) | | | 0.00 [0.00, 0.01] p=<0.01 *** | | | |
| Grocery (lag 2) | | | | 0.00 [−0.00, 0.01] p=0.08 + | | |
| Retail (lag 2) | | | | | −0.00 [−0.00, 0.00] p=0.65 | |
| Residence (lag 2) | | | | | | −0.02 [−0.03, −0.01] p=<0.01 *** |
| Num.Obs. | 1632 | 1626 | 1622 | 1630 | 1628 | 1632 |
| R2 Adj. | 0.957 | 0.957 | 0.957 | 0.957 | 0.957 | 0.957 |

The numbers show the unstandardized coefficients from linear regression models, which indicate the implied change on the log of the number of COVID-19 cases for a one-unit change in the covariate. The models include indicators at the province level (N = 12), as well as controls for the demographic structure of the provinces (share of 65 +, share of women, share of low-income households, share of 1st generation immigrants). 95% Confidence intervals are reported in the square brackets. Significance levels of p values: *** < 0.001; ** < 0.01; * < 0.05; + < 0.10. The precise p values are printed when > 0.01.

significant negative associations with excess mortality. In terms of effect size, the one-step ahead direct effect of the imposition of some restrictive measures (one point on the 1-to-10 scale) is a reduction of excess mortality with approximately 2.5 percentage points (e.g., from the median of 4.7% to 2.2% excess mortality).

Importantly, the effect of policy remains significant although in declines in size when the mobility measures are included in the models. This implies that mobility changes do not fully mediate the effect of policy. The biggest reduction in the size of the coefficient (approximately 27%) is observed when mobility related to transport and presence in residential places is included. The effect of mobility in the presence of policy, however, is not precisely estimated and it sensitive the lag structure. This has to do to a large extent with the collinearity (contemporarious correlations) between policy stringency and mobility measures.

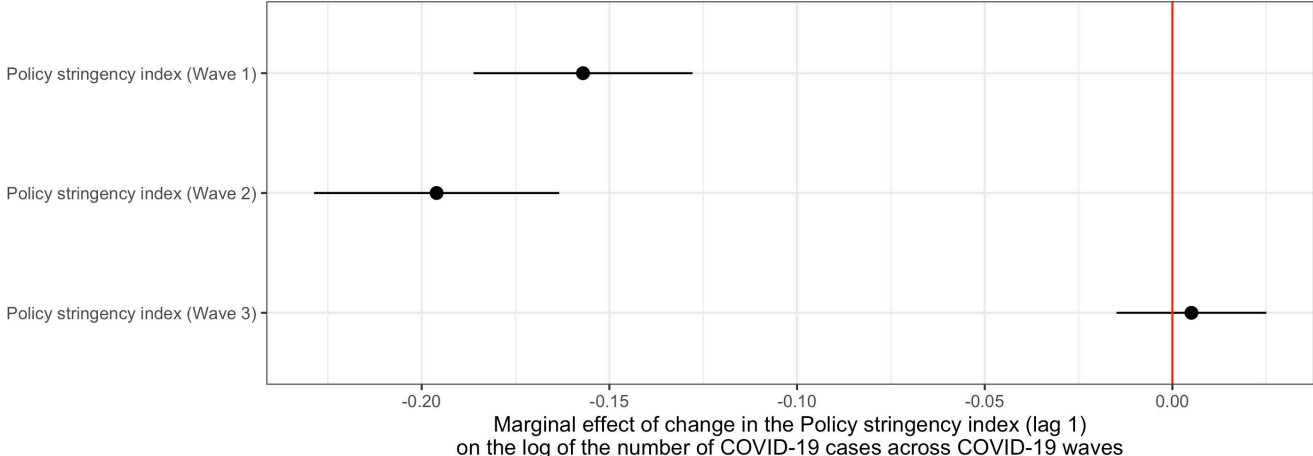

**Fig 2. Marginal effects of changes in the Policy stringency index (lag 1) on the log of the number of COVID-19 cases in The Netherlands, across the first three waves of the pandemic (2020-2022).** The figure shows the predicted marginal effects (point estimates and 95% confidence intervals) of one-point increases in the Policy stringency index (lagged with one week) on the log of the number of COVID-19 for the three waves of the pandemic in The Netherlands, 2020–2022. Results based on Model S3a in S3 Table in S1 File.

When we explore the effect of mobility while excluding the policy stringency measure, we observe the expected effects of mobility, so that excess mortality decreases significantly with lower presence in places of work, transport, grocery and retail, and with longer presences in residential places (see Fig 3). In addition, the precise lag with which the mobility measures are included does not make a big difference. The figure also shows that the effect of policy stringency is consistent across different lags between 4 and 7.

From the control variables, it is noteworthy that colder weeks (with weekly average of minimum temperature lower than 0°C) have a significant positive association with excess mortality, while very warm weeks (with weekly average of daily maximum temperature above 25°C) do not. Public holidays are associated with higher excess mortality, even though they were not associated with more COVID-19 cases, which might be related to reduced capacity to deal with holiday-related incidences.

When we look at possible interaction effects, the policy effect on excess mortality is most pronounced during the first wave and gets smaller during the second and third waves (for details see S4 Table in S1 File). The effect was greatest in the Southern provinces Limburg and North Brabant (see S4 Table in S1 File and Fig 4) and it was smallest in the less densely populated provinces Drenthe, Flevoland and Zeeland.

The results remain similar when we model the number of registered COVID-19 deaths instead of our estimates of excess mortality (see S5 Table in S1 File): policy stringency continues to have a significant negative association with the number of COVID-19 deaths five weeks ahead, and the pattern of results for the effects of mobility is very similar to the one in Table 3.

The results are robust to including indicators for the calendar months. In fact, many of the coefficients of interest are estimated more precisely and the policy-related reductions appear slightly bigger in size.

## Conclusions

The study examined the impact of COVID-19 policy restrictions on excess mortality in The Netherlands for the 2020−2022 period. We theorized the mechanisms connecting COVID-19 cases and deaths, policy stringency, changes in mobility patterns, and excess mortality and modeled the relationships among these using data disaggregated weekly and at the province level. The focus on excess mortality in place of COVID-19 cases and related deaths alone [17,21], its analysis

**Table 3. Linear regression models of excess mortality (the log of IRR, or the Incidence Rate Ratio).**

| | Model 3a log(IRR) | Model 3b log(IRR) | Model 3c log(IRR) | Model 3d log(IRR) | Model 3e log(IRR) | Model 3f log(IRR) |
|---|---|---|---|---|---|---|
| Public holidays | 0.215 [0.181, 0.249] p < 0.001 *** | 0.206 [0.172, 0.239] p < 0.001 *** | 0.216 [0.183, 0.249] p < 0.001 *** | 0.215 [0.181, 0.248] p < 0.001 *** | 0.215 [0.181, 0.249] p < 0.001 *** | 0.216 [0.182, 0.249] p < 0.001 *** |
| Weekly average of daily minimum temperature < 0 | 0.017 [0.008, 0.026] p < 0.001 *** | 0.020 [0.011, 0.029] p < 0.001 *** | 0.017 [0.009, 0.026] p < 0.001 *** | 0.016 [0.007, 0.024] p < 0.001 *** | 0.016 [0.007, 0.025] p < 0.001 *** | 0.018 [0.009, 0.027] p < 0.001 *** |
| Weekly average of daily maximum temperature > 25 | −0.002 [−0.012, 0.008] p = 0.638 | −0.003 [−0.013, 0.007] p = 0.595 | −0.001 [−0.011, 0.009] p = 0.808 | −0.002 [−0.011, 0.008] p = 0.765 | −0.002 [−0.012, 0.008] p = 0.745 | −0.002 [−0.012, 0.008] p = 0.671 |
| Policy stringency index (lag 5) | −0.022 [−0.026, −0.019] p < 0.001 *** | −0.018 [−0.022, −0.014] p < 0.001 *** | −0.016 [−0.022, −0.010] p < 0.001 *** | −0.023 [−0.028, −0.019] p < 0.001 *** | −0.024 [−0.029, −0.018] p < 0.001 *** | −0.016 [−0.022, −0.009] p < 0.001 *** |
| Work (lag 6) | | 0.002 [0.001, 0.003] p < 0.001 *** | | | | |
| Transport (lag 6) | | | 0.001 [0.000, 0.002] p = 0.005 ** | | | |
| Grocery (lag 6) | | | | −0.001 [−0.002, 0.000] p = 0.128 | | |
| Retail (lag 6) | | | | | −0.000 [−0.001, 0.000] p = 0.419 | |
| Residence (lag 6) | | | | | | −0.004 [−0.008, −0.000] p = 0.030 * |
| Num.Obs. | 1596 | 1578 | 1574 | 1582 | 1580 | 1584 |
| R2 Adj. | 0.174 | 0.178 | 0.174 | 0.171 | 0.170 | 0.171 |

The numbers show the unstandardized coefficients from linear regression models, which indicate the implied change on excess mortality (defined as the log of IRR, or Incidence Rate Ratio) for a one-unit change in the covariate. The models include indicators at the province level (N = 12), as well as controls for the demographic structure of the provinces (share of 65+, share of women, share of low-income households, share of 1st generation immigrants). 95% Confidence intervals are reported in the square brackets. Significance levels of p values: *** < 0.001; ** < 0.01; * < 0.05; + < 0.10. The precise p values are printed when > 0.01.

beyond the first wave to include later stages of the pandemic, and the inclusion of within-country variation [13,14] together mark the importance of our contribution.

We show that the number of COVID-19 cases and deaths is associated with the strictness of policy restrictions, which is no surprise, as policy makers were actively monitoring and responding to the information about the course of the pandemic. It is nevertheless informative to see a confirmation of this in the data, and the differential impact of the cases and deaths over the course of the pandemic, in particular.

Our results confirm the findings of previous research that policy restrictions, partly by reducing mobility, were associated with curbing the spread of COVID-19 and limiting excess mortality [10]. We found that greater policy stringency led to reductions in presence at workplaces, transportation hubs, groceries and retail locations, while increasing presence at home. These changes in mobility patterns in turn were associated with lower COVID-19 case counts two weeks later. Ultimately, more stringent policies were followed by lower excess mortality with a lag of six weeks. Some, but not all of the policy-related reductions, were, exercised by limiting mobility in areas with a potential for high transmission.

Importantly, we observed significant variation in these relationships across the different waves of the pandemic. The impact of case numbers on the stringency of restrictions declined with each wave, while that of deaths was constant in the

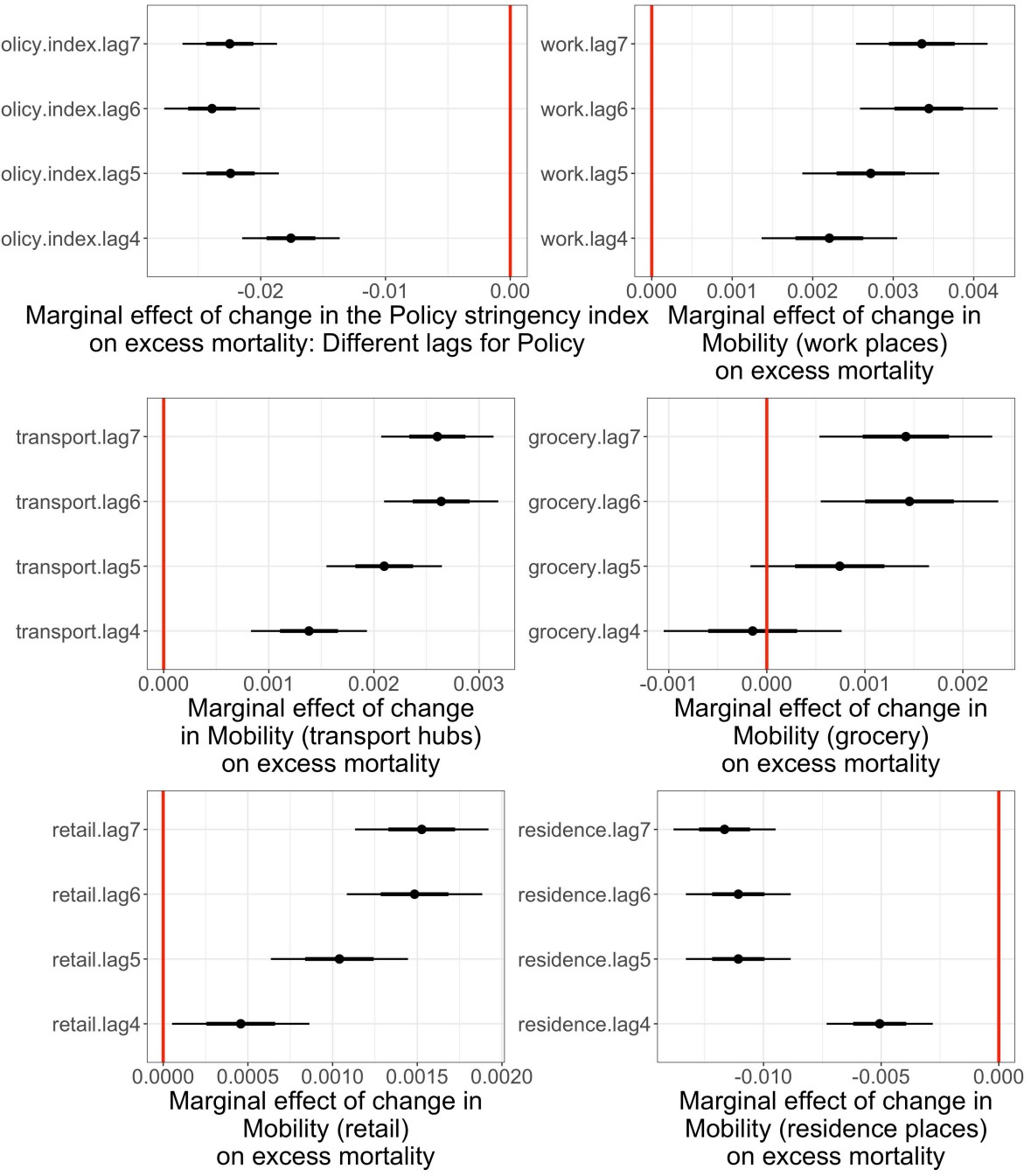

**Fig 3. Marginal effects of Policy stringency and Mobility on excess mortality (log IRR) for different lags of the predictors, in The Netherlands, 2020-2022.** The figure shows the predicted marginal effects (point estimates and 95% confidence intervals) of one-point increases in the Policy stringency index (top left panel) and different aspects of Mobility (other five panels) on excess mortality (defined as the log of the IRR, Incidence Rate Ratio) in The Netherlands, 2020–2022, for different lags of the predictors. The coefficients for mobility are from models similar to the ones reported in Table 3, but with policy stringency is excluded.

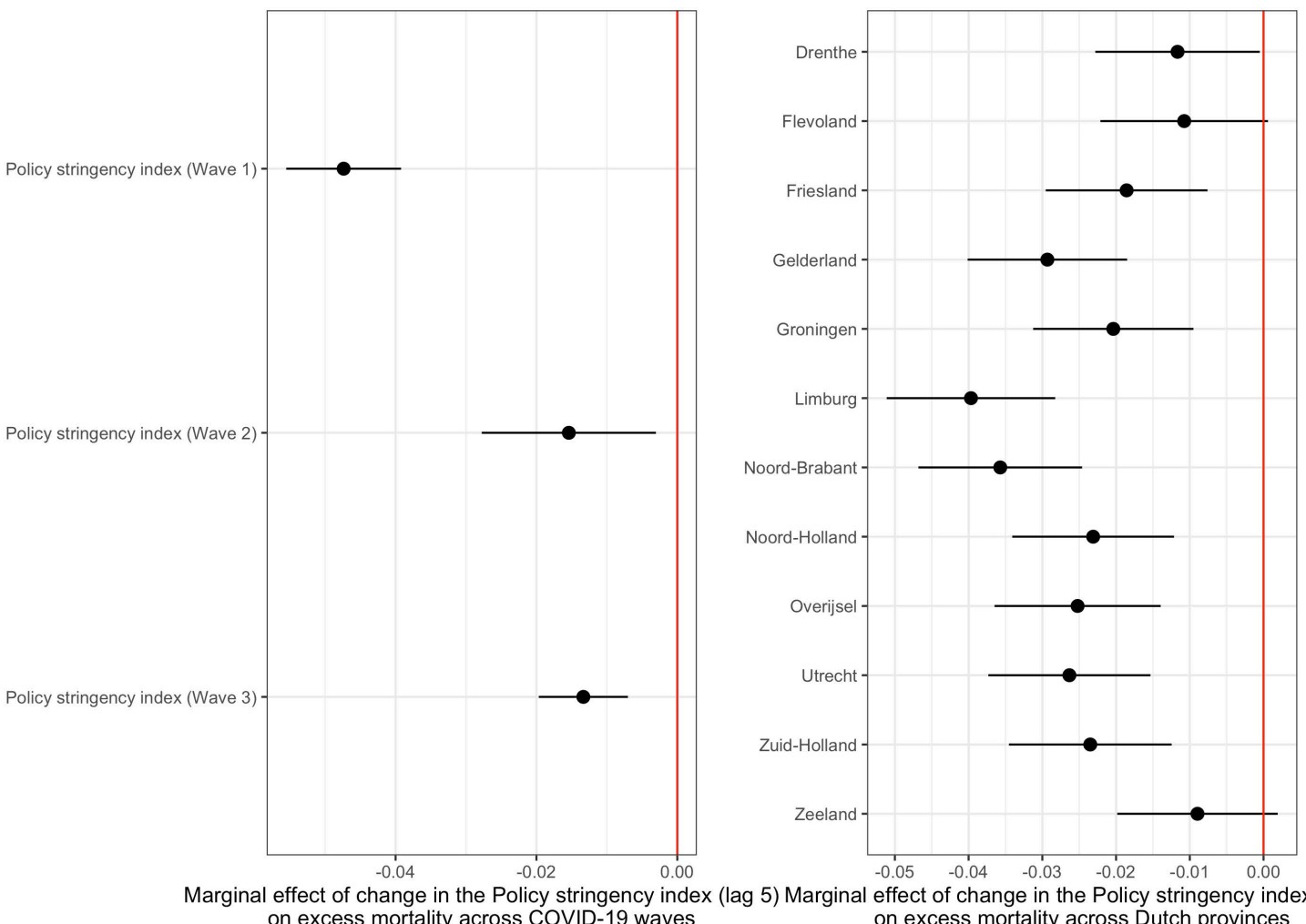

**Fig 4. Marginal effects of changes in the Policy stringency index (lag 5) on excess mortality (log IRR) across COVID-19 waves (left panel) and across Dutch provinces (right panel), 2020-2022.** The figure shows the predicted marginal effects ((point estimates and 95% confidence intervals) of one-point increases in the Policy stringency index (lagged with five weeks) on the log of the IRR (Incidence Rate Ratio) for the three waves of the pandemic (left panel) and for the 12 provinces (right panel) in The Netherlands, 2020–2022. Results based on the models reported in S4 Table in S1 File.

first two waves but increased in the third. Policymakers may have become more responsive to the most severe health outcomes as the pandemic continued, as the virus variant during the third wave was milder than the previous variants [30]. The effect of policy stringency on mobility was strongest during the first wave and declined in subsequent waves. This may reflect 'pandemic fatigue' and reduced compliance over time. Relatedly, the impact of policy on the number of cases was most pronounced in the second wave. The smaller policy-related reductions during the third wave could be due to the progress of the COVID-19 vaccination campaign.

Interestingly, we did not find a significant relationship between changes in presence in retail and recreation establishments and COVID-19 case counts. This contrasts with the strong effects observed for workplace and transportation mobility. It may be that other preventive behaviors, such as mask-wearing (which became officially recommended during the second wave in the fall of 2020) and distancing, were important for reducing transmission in these essential retail settings. While we cannot rule out all sources of confounding due to the endogenous nature of policy adoption, our conditional

association framework provides valuable evidence for policy deliberation. Overall, our findings highlight the critical role that policy played in mitigating the health impacts of COVID-19, via redactions in mobility and other channels. However, the declining effectiveness of these measures over time, and the existence of direct policy-related reductions beyond mobility, underscore the need for policymakers to continually assess and adapt their strategies as pandemics evolve. At the same time, policymakers must carefully weigh the benefits of restrictive measures against the significant costs they impose on personal liberty.

Our study has some important limitations. Having cross-sectional variation in policy restrictions, in addition to the variation over time, would provide additional leverage for the causal identification of policy effects. Furthermore, our approach is not able to estimate the effects of individual policy measures, such as school closures, because the restrictions came in packages that bundled different individual policy measures together. Lastly, the equations we estimate need to be embedded into an integrated dynamic model of pandemic development so that an assessment of the total *cumulative* impact of the policy measures can be made (cf. [2]).

Further research into the effectiveness of different policy mixes is essential to guide pandemic responses and prepare for future public health emergencies. For example, it would be fruitful to see replications of our analytic approach in other countries with more regional variation in policy restrictions and in a cross-country time-series analysis. Communicating the findings of such research so that it can help decision makers strike the necessary balance between public health objectives and individual rights is equally important. Responses to the next pandemic can benefit from valuables lessons learned from the experience of COVID-19 with respect to the effectiveness of government interventions and the limits to their effects.

## Supporting information

**S1 File. The Impact of Policy Restrictions PLOS SI.pdf.** Additional analyses and robustness checks.
(PDF)

## Acknowledgments

We would like to thank the members of the research team part of the project on excess mortality in The Netherlands (see S7 Table in S1 File).

## Author contributions

**Conceptualization:** Dimiter Toshkov, Camila Caram-Deelder, Brendan Carroll, Frits Rosendaal.

**Data curation:** Dimiter Toshkov, Camila Caram-Deelder.

**Formal analysis:** Dimiter Toshkov.

**Funding acquisition:** Camila Caram-Deelder, Frits Rosendaal.

**Investigation:** Dimiter Toshkov.

**Methodology:** Dimiter Toshkov, Camila Caram-Deelder, Brendan Carroll, Frits Rosendaal.

**Project administration:** Camila Caram-Deelder, Frits Rosendaal.

**Resources:** Frits Rosendaal.

**Supervision:** Frits Rosendaal.

**Validation:** Brendan Carroll.

**Visualization:** Dimiter Toshkov.

**Writing – original draft:** Dimiter Toshkov.

**Writing – review & editing:** Dimiter Toshkov, Camila Caram-Deelder, Brendan Carroll, Frits Rosendaal.

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
