## [Decision Letter · Decision Letter 0]

19 May 2025

Dear Dr. Toshkov,

Thank you for submitting your manuscript to PLOS ONE. After careful consideration, we feel that it has merit but does not fully meet PLOS ONE’s publication criteria as it currently stands. Therefore, we invite you to submit a revised version of the manuscript that addresses the points raised during the review process.

We look forward to receiving your revised manuscript.

Kind regards,

Flavio A. Ziegelmann, Ph.D.

Academic Editor

PLOS ONE

Journal Requirements:

2. Thank you for stating the following in the Acknowledgments Section of your manuscript: [We would like to thank the members of the research team part of the project on excess mortality in The Netherlands financed by the Dutch organization for knowledge and innovation in health, healthcare and well-being (ZomMW): No 104.302.522.100.07 and 104.302.522.200.03..]

Please remove any funding-related text from the manuscript and let us know how you would like to update your Funding Statement. Currently, your Funding Statement reads as follows: [FR and CCD received funding from the Dutch organization for knowledge and innovation in health, healthcare and well-being (ZomMW): No 104.302.522.100.07 and 104.302.522.200.03. The funder did not play any role in the study design, data collection and analysis, decision to publish, or preparation of the manuscript.]

3, When completing the data availability statement of the submission form, you indicated that you will make your data available on acceptance. We strongly recommend all authors decide on a data sharing plan before acceptance, as the process can be lengthy and hold up publication timelines. Please note that, though access restrictions are acceptable now, your entire data will need to be made freely accessible if your manuscript is accepted for publication. This policy applies to all data except where public deposition would breach compliance with the protocol approved by your research ethics board. If you are unable to adhere to our open data policy, please kindly revise your statement to explain your reasoning and we will seek the editor's input on an exemption. Please be assured that, once you have provided your new statement, the assessment of your exemption will not hold up the peer review process.

Additional Editor Comments:

The paper addresses an interesting and still open applied problem. It is clear and well written. Nevertheless, I agree with the reviewers, particularly reviewer 2, that some important points need to be dealt with. So, I hope the authors can provide a fully revised version changed according to their criticisms and sugestions.

Reviewers' comments:

Reviewer's Responses to Questions

**Comments to the Author**

1. Is the manuscript technically sound, and do the data support the conclusions?

Reviewer #1: Yes

Reviewer #2: Yes

Reviewer #3: Yes

2. Has the statistical analysis been performed appropriately and rigorously?

Reviewer #1: Yes

Reviewer #2: No

Reviewer #3: Yes

3. Have the authors made all data underlying the findings in their manuscript fully available?

Reviewer #1: Yes

Reviewer #2: Yes

Reviewer #3: No

4. Is the manuscript presented in an intelligible fashion and written in standard English?

Reviewer #1: Yes

Reviewer #2: Yes

Reviewer #3: Yes

Reviewer #1: This is an automated report for PONE-D-25-14743. This report was solicited by the PLOS One editorial team and provided by ScreenIT.

ScreenIT is an independent group of scientists developing automated tools that analyze academic papers. A set of automated tools screened your submitted manuscript and provided the report below. Each tool was created by your academic colleagues with the goal of helping authors. The tools look for factors that are important for transparency, rigor and reproducibility, and we hope that the report might help you to improve reporting in your manuscript. Within the report you will find links to more information about the items that the tools check. These links include helpful papers, websites, or videos that explain why the item is important. While our screening tools aim to improve and maintain quality standards they may, on occasion, miss nuances specific to your study type or flag something incorrectly. Each tool has limitations that are described on the ScreenIT website. The tools screen the main file for the paper; they are not able to screen supplements stored in separate files. Please note that the Academic Editor had access to these comments while making a decision on your manuscript. The Academic Editor may ask that issues flagged in this report be addressed. If you would like to provide feedback on the ScreenIT tool, please email the team at ScreenIt@bih-charite.de. If you have questions or concerns about the review process, please contact the PLOS One office at plosone@plos.org.

Reviewer #2: The study's merit lies in its ability to provide concrete causal estimations of the size and impacts of stringency policies during three waves of COVID-19 pandemics. For that, authors rely on panel data, with observations from different regions within a single country. In my interpretation, if this is done in a compelling manner, according to reviewers' guidelines, the paper would be suitable for publication in PLOS ONE, as an original research, given the current state of literature, far from reaching a final conclusion on the topic.

What remains to be assessed, therefore, is the soundness of the identification strategy used. At this point, I am concerned the paper, in the current form, is not suitable for publication, as many parameters and assumptions are made throughout the text, and left unexplained, at the expense of a compelling argument about the plausibility of the causal claims.

It is my interpretation that the paper should make a more compelling argument on the necessary conditions for causality and how they are met, relying on the existing literature, to convince readers that the strategy is as clean as possible and overcomes any possible threats, specially endogeneity coming from this circular causality framework. I am not completely aware of the literature's state of the art on this instance, but the authors should elaborate on how considering the lags of variables solves the problem, and how to put together the three different regression results to make one compelling argument.

Also, I am very concerned about the excess use of parameters and assumptions at the author's discretion with no proper justification or robustness tests. Examples are: (i) rescaling of the stringency index; (ii) the definition of 'Information' variable using log(cases/1.000) on one hand, and new COVID-19 deaths in level (and lagged) at the other; (iii) the definition of what Google Mobility variables enter the study; (iv) definitions of lags being 1 week at first, then 6 weeks, then 5 weeks, all in the main specifications.

Finally, for the first regression, presented in Table 1, besides the fact that it is actually not used in the study's argument, I have another comment: it is perhaps not clear, but it is my understanding that any observations $stringency_{{i,t}}$ and $stringency_{j,t}$ are exactly the same for any pair $(i,j)$ of provinces, as stringency is measured at the country level. If this is the case, I am not sure the regression makes any sense at all. In fact, I'd assume policy makers evaluate cases and death numbers at the national level (i.e. simultaneously assessing all provinces' outcomes) to define a nation-wide policy. What type of causal claim can be made here with that premise in mind and considering there is no variation at the province level?

Reviewer #3: Abstract:

The article analyzes the impact of COVID-19 policy restrictions on mobility patterns and excess mortality at the regional level in the Netherlands between 2020 and 2022. It combines data on public policies, mobility patterns from Google Mobility Reports, officially recorded COVID-19 cases and deaths, and region-specific measures of excess mortality over time. Among the main findings, the authors find that the stringency of policy restrictions increased with the number and growth rates of COVID-19 cases and deaths. Public mobility decreased, while presence in residential locations increased in response to stricter policies and rising COVID-19 case and death counts in previous weeks. Furthermore, excess mortality decreased with stricter policy restrictions. In sum, the political restrictions were effective in limiting the spread of the pandemic and in saving lives.

Comments:

• The article satisfies the criteria outlined above; nevertheless, several points require further consideration before it can be published.

• Overall, the text is well written, presenting a clear and well-developed argument throughout. In the introduction, when referencing the literature, it would be helpful to briefly outline the contributions of the cited works, to ensure the references are not perceived as vague or lacking context.

• Using lagged effects—where mobility and policies respond immediately to changes in the COVID-19 growth rate, but influence cases only after a certain time period—contributes to mitigating the endogeneity problem. However, the system of equations is ultimately estimated in isolation, without incorporating a selection bias correction term or an instrumental variable. Instead, separate equations are estimated using lags to capture the effects. It would strengthen the econometric approach to establish a more cohesive relationship among these elements.

• The first part of the results explains restriction policies as a function of reported COVID-19 cases and deaths. However, it is not entirely clear how the results from the first part are incorporated into the second. If mobility depends on policy, and policy is determined by cases and deaths, there should be some temporal structure that allows these effects to be observed. It would be important to clarify how the use of lags fully addresses this issue. A first-stage estimation might be necessary to properly incorporate these effects into the model.

• What explains the variation in the coefficients of the change in mobility with respect to excess mortality when the number of lag weeks is altered? A discussion of the results is crucial for understanding the optimal duration of the policy.

**Do you want your identity to be public for this peer review?** For information about this choice, including consent withdrawal, please see our Privacy Policy

Reviewer #1: No

Reviewer #2: **Yes:** Arthur Sonntag Kuchenbecker

Reviewer #3: No

---

## [Decision Letter · Decision Letter 1]

15 Sep 2025

Dear Dr. Toshkov,

Thank you for submitting your manuscript to PLOS ONE. After careful consideration, we feel that it has merit but does not fully meet PLOS ONE’s publication criteria as it currently stands. Therefore, we invite you to submit a revised version of the manuscript that addresses the points raised during the review process.

We look forward to receiving your revised manuscript.

Kind regards,

Flavio A. Ziegelmann, Ph.D.

Academic Editor

PLOS ONE

Journal Requirements:

Additional Editor Comments:

Reviewer #2:

Reviewer #3:

Reviewers' comments:

Reviewer's Responses to Questions

**Comments to the Author**

Reviewer #2: All comments have been addressed

Reviewer #3: (No Response)

2. Is the manuscript technically sound, and do the data support the conclusions?

Reviewer #2: Partly

Reviewer #3: Partly

3. Has the statistical analysis been performed appropriately and rigorously?

Reviewer #2: Yes

Reviewer #3: N/A

4. Have the authors made all data underlying the findings in their manuscript fully available?

Reviewer #2: Yes

Reviewer #3: Yes

5. Is the manuscript presented in an intelligible fashion and written in standard English?

Reviewer #2: Yes

Reviewer #3: Yes

Reviewer #2: It is my understanding that the paper has 3 main threats to validity, that are now fully transparent and clearly explained by the authors in the conclusion and empirical approach sections. I outline them in my attached file, in my terms, and leave for the Editor the judgment if it's fit for the journal with these possible gaps in mind.

Reviewer #3: The article shows progress compared to the previous version, with the incorporation of several reviewers’ suggestions. The discussion on transmission mechanisms was expanded, the treatment of lags was more detailed, and the relationship between policy, mobility, cases, and mortality became clearer. The limitations were also acknowledged, recognizing restrictions related to the spatial variation of policies, the difficulty of isolating specific measures, and the absence of an integrated dynamic model.

However, the main weakness remains the issue of endogeneity. The problem of reverse causality between cases/deaths and policy adoption persists, as well as the simultaneity between mobility and restrictions, particularly between the evolution of COVID-19 and the policy index. The use of temporal lags is a defensible strategy and consistent with the literature, but it does not eliminate the risk of bias, since lagged COVID variables may themselves explain the adoption of measures. Thus, the study resembles more an empirical analysis of conditional correlations than an identified estimation of causal effects. This limitation undermines the strength of the evidence, and I do not find it sufficient to recommend publication at this stage.

**Do you want your identity to be public for this peer review?** For information about this choice, including consent withdrawal, please see our Privacy Policy

Reviewer #2: No

Reviewer #3: No

---

## [Decision Letter · Decision Letter 2]

11 Dec 2025

The Impact of Policy Restrictions and Mobility Changes on Excess Mortality During the COVID-19 Pandemic in The Netherlands, 2020-2022

PONE-D-25-14743R2

Dear Dr. Toshkov,

We’re pleased to inform you that your manuscript has been judged scientifically suitable for publication and will be formally accepted for publication once it meets all outstanding technical requirements.

Kind regards,

Flavio A. Ziegelmann, Ph.D.

Academic Editor

PLOS One

Additional Editor Comments (optional):

I am happy to say that I agree with the referees' comments about the improvement of the third version of the paper, despite the remaining pitfalls raised by both. I am therefore favourable to accept the manuscript with very minor modifications suggested by one of the referees. It does not need to be further evaluated though.

Reviewers' comments:

Reviewer's Responses to Questions

**Comments to the Author**

Reviewer #2: All comments have been addressed

Reviewer #3: All comments have been addressed

2. Is the manuscript technically sound, and do the data support the conclusions?

Reviewer #2: Partly

Reviewer #3: Yes

3. Has the statistical analysis been performed appropriately and rigorously?

Reviewer #2: Yes

Reviewer #3: I Don't Know

4. Have the authors made all data underlying the findings in their manuscript fully available?

Reviewer #2: Yes

Reviewer #3: Yes

5. Is the manuscript presented in an intelligible fashion and written in standard English?

Reviewer #2: Yes

Reviewer #3: Yes

Reviewer #2: I'm satisfied with the answers and clarity on threats to validity in the text. I acknowledge the setting is complex for causal estimation, and that although impossible to make a final claim on causality, results are relevant for policy evaluation.

Reviewer #3: The article “The Impact of Policy Restrictions and Mobility Changes on Excess Mortality During the COVID-19 Pandemic in The Netherlands, 2020–2022” has advanced in relation to the previous versions by further developing the observational panel design and explaining the lags in the policy, mobility, and COVID-19 case variables in the main equations. The inclusion of these lags helps to mitigate, albeit only partially, problems of simultaneity and endogeneity between policies, mobility, cases, and excess mortality, making the temporal dynamics more consistent with the epidemiological process and with policy responses. Nevertheless, the identification is fundamentally correlational, so the coefficients should be interpreted as conditional associations and not as strict causal effects.

I understand, however, that the adjustments made are sufficient for the scope of the article and the journal. I would only recommend, as a minor presentational adjustment, simplifying the result tables, keeping mainly the estimated coefficient, the standard error, and the indication of statistical significance via asterisks, without the need to report p-values and full confidence intervals simultaneously, as this tends to overload the reading. Finally, I would reinforce in the conclusion the need to translate the results more directly into implications for the post-pandemic period, making the study’s contribution clear to the reader and thereby increasing its usefulness for the public policy debate in future events.

**Do you want your identity to be public for this peer review?** For information about this choice, including consent withdrawal, please see our Privacy Policy

Reviewer #2: **Yes:** Arthur Sonntag Kuchenbecker

Reviewer #3: No

---

## [Editor Report · Acceptance letter]

PONE-D-25-14743R2

PLOS One

Dear Dr. Toshkov,

I'm pleased to inform you that your manuscript has been deemed suitable for publication in PLOS One. Congratulations! Your manuscript is now being handed over to our production team.

Kind regards,

on behalf of

Professor Flavio A. Ziegelmann

Academic Editor

PLOS One